# Copper-Doped Bioactive Glass/Poly (Ether-Ether-Ketone) Composite as an Orbital Enucleation Implant in a Rabbit Model: An In Vivo Study

**DOI:** 10.3390/ma15134410

**Published:** 2022-06-22

**Authors:** Ke Xiong, Mengen Zhao, Zhaoying Wu, Wei Zhang, Chao Zhang

**Affiliations:** 1School of Biomedical Engineering, Shenzhen Campus, Sun Yat-sen University, Shenzhen 518107, China; xiongk3@mail2.sysu.edu.cn (K.X.); zhaomen@mail2.sysu.edu.cn (M.Z.); wuzhy37@mail.sysu.edu.cn (Z.W.); 2Department of Ophthalmology, Nanfang Hospital, Southern Medical University, Guangzhou 510515, China; 3Department of Outpatient, The First Affiliated Hospital, Sun Yat-sen University, Guangzhou 510080, China

**Keywords:** bioactive glass, poly(ether-ether-ketone), scaffold, enucleation, orbital implant

## Abstract

An orbital enucleation implant is used to compensate for the orbital volume deficits in the absence of the globe. In this work, copper-doped bioactive glass in poly(ether-ether-ketone) (CuBG/PEEK) composite scaffolds as an orbital enucleation implant were designed and fabricated by cool-pressed sintering and particle-leaching techniques, the incorporation of copper-doped bioactive glass in poly(ether-ether-ketone) (CuBG/PEEK) was expected to significantly improve the biocompatibility of the PEEK implant. The consequences after implantation of the CuBG/PEEK composite scaffolds in experimental, eviscerated rabbits was observed and assayed in term of histopathological examination. In detail, 24 rabbits were randomly divided into three groups: Group A, PEEK scaffolds; Group B, 20% CuBG/PEEK composite scaffolds; Group C, 40% CuBG/PEEK composite scaffolds; the rabbits were sacrificed at week 4 and week 12, followed by histochemical staining and observation. As a result, the PEEK group exhibited poor material exposure and tissue healing, while the CuBG/PEEK scaffolds showed good biocompatibility, and the 40% CuBG/PEEK composite scaffold exhibited the best performance in angiogenesis and tissue repair. Therefore, this study demonstrates the potential of CuBG/PEEK composite scaffolds as an orbital enucleation implant.

## 1. Introduction

When an eye has become irreversibly blind or undergone severe injury/tumor, it is removed by enucleation or evisceration to control pain or alleviate the infection. Following the removal of the eye, an orbital implant is inserted into the ophthalmic socket in order to provide satisfactory volume replacement and restore the aesthetic appearance of a normal eye. An orbital implant can compensate for the orbital volume deficits in the absence of the globe [1,2,3]. Over the past two centuries, an extensive variety of materials has been used to fabricate orbital implants, some of which even resulted in disastrous results.

The unique structure of porous materials used as an enucleation implant allows vascular and tissue ingrowth and, in turn, helps to anchor the implant and permits immune surveillance [4]. Since their introduction as enucleation implants in the late 1980s, porous materials have become widely used in clinical practice [5,6]. These would include hydroxyapatite [7], high-density polyethylene [8], aluminum oxide [9], bone cement [10], etc. Despite the initial success of porous implants and reports of low extrusion rates, a number of problems, such as the risk of infection, the development of late exposures, and the formation of pyogenic granuloma, remain unsolved, and there is a growing urge for biocompatible orbital implants [11,12]. An ideal material for orbital enucleation should better possess a similar density/weight to the natural globe, proper porosity, appreciable histocompatibility, and cost-effectiveness, and is expected to achieve minimal rates of migration, extrusion, exposure, and infection [13]. Poly(ether-ether-ketone) (PEEK) is a semicrystalline, thermoplastic polymer, and it is usually synthesized by *Friedel-Crafts* polycondensation of 4,4′-difluorobenzophenone with disodium salt of hydroquinone [14]. As a high-performance polymer, PEEK is known for its excellent mechanical properties, as well as extraordinary thermal stability and chemical resistance against oils, acids, and biological fluids [14]. With a continuous use temperature of 260 °C, PEEK is suitable for most clinical sterilization techniques; moreover, it did not cause artifacts in computed tomography (CT) images [15]. Because of its excellent biocompatibility in vitro and in vivo, PEEK is already used for long-term medical implant applications. However, the biological inertness of PEEK has hindered its wide application. Therefore, it is highly desirable to enhance the bioactivity of PEEK via the introduction of bioactive ingredients/components to the PEEK matrix [16].

As an eminent biomaterial for bone repair/regeneration, bioactive glass (BG) is known for its superior bioactivity and biocompatibility and has been used in clinical practice [17,18,19]. The introduction of bioactive materials in PEEK can significantly improve its biocompatibility, making it more suitable for orbital implant. Copper is involved in the angiogenesis process [20,21,22], and copper ions are known to not only improve the anti-infective ability of biomedical materials, but also to induce the proliferation of endothelial cells, and blood vessel formation mainly depends on the activity of endothelial cells [20,21,22].

In the present study, we aimed to design a PEEK-based implant material with enhanced bioactivity and evaluate its applicability as the orbital implant. To this end, copper-doped bioactive glass nanoparticles (CuBG) were prepared and incorporated into the PEEK matrix to fabricate CuBG/PEEK composite scaffolds; the scaffolds were implanted into experimental, eviscerated rabbits to observe the consequences and histopathological changes after implantation. Such an investigation will help establish a substantial foundation for the design and manufacture of new orbital implants with multifunctional properties.

## 2. Materials and Methods

### 2.1. Materials

PEEK powder (99%, Junhua PEEK, Changzhou, China), tetraethyl orthosilicate (TEOS, Tianjin Zhiyuan Reagent, Tianjin, China), ammonium hydroxide (NH_3_·H_2_O, 28%, Tianjin Zhiyuan Reagent, Tianjin, China), calcium nitrate tetrahydrate (Ca(NO_3_)_2_·4H_2_O, 99%, Aladdin, Shanghai, China), copper tetrahydrate (Cu(NO_3_)_2_·3H_2_O, 98%, Macklin, Shanghai, China), paraformaldehyde solution (4%, Seville Creature, Beijing, China), hematoxylin/eosin (H&E) staining kit (Solarbio, Beijing, China), and Masson’s trichrome staining solution (Solarbio, Beijing, China) were used as received. All other solvents were of analytical grade and used without purification.

### 2.2. Synthesis of CuBG

CuBG was synthesized through the typical sol–gel method according to previous studies [23]. The solution, containing 3.6 mL of tetraethyl orthosilicate (TEOS, >99%), 7 mL of ammonium hydroxide, and 33 mL of deionized water, was placed in a constant-temperature water tank at 35 °C. The mixture was allowed to react for 4 h after the addition of 1.64 g of calcium nitrate tetrahydrate and 0.56 g copper nitrate trihydrate. The suspension was centrifuged at 8000 rpm (Centrifuge 5430R, Eppendorf, Hamburg, Germany) for 10 min to collect deposits, which were further washed twice with ethanol and once with deionized water. Afterwards, the deposits were dried at 60 °C for 24 h before calcination at 600 °C for 3 h in a muffle furnace (heating rate of 2 °C/min).

### 2.3. Fabrication of CuBG/PEEK Scaffolds

The CuBG/PEEK composite scaffolds were prepared using the cool-pressed sintering and particle-leaching method [24]. Briefly, the PEEK powder and CuBG powder were mixed at a certain ratio (80:20, 60:40), and ball-milled to obtain mixture powders. Then, a predetermined quantity of sodium chloride particles (with a particle size of 400–500 μm) was added into the mixture powders at a weight ratio of 1:8; the mixture was then transferred to a stainless-steel mold (Φ: 12 mm) and pressed under 20 MPa for 8 min at room temperature. The specimens were subjected to sintering in a furnace at 345 °C for 2 h, and then rinsed in deionized water for 72 h to dissolve the NaCl particles. The obtained scaffolds were then dried at 37 °C for 24 h and named CP20 and CP40 according to the weight percentage of CuBG to PEEK. PEEK scaffolds (CP0) were prepared by the same process as the control.

### 2.4. XRD Analysis

The phase composition and structural characteristics of the CuBG/PEEK composite scaffolds were characterized by X-ray diffraction (XRD, Empyrean, Panaco, Almelo, The Netherlands) in a 2θ range of 10–80° and Fourier transform infrared spectrometry (FTIR, NICOLET 6700, Madison, WI, USA) using the KBr pellet method in a region between 2400 and 400 cm^−1^, with a resolution of 4 cm^−1^.

### 2.5. Morphological Study

The surface morphology and composition of CuBG/PEEK composite scaffolds (n = 3 per group) were observed by field-emission scanning electron microscopy (FESEM, Quanta 400F, FEI, Hillsboro, OR, USA) and energy-dispersive spectrometry (EDS, Quanta 400F, FEI, Hillsboro, OR, USA).

### 2.6. Porosity Measurement

The porosity (P) of the CuBG/PEEK composite scaffolds was calculated according to Archimedes’ principle via the use of a gravity bottle. Briefly, the dry mass of the scaffold (M_d_) was recorded. Then, the scaffold was soaked in cyclohexane in a specific-gravity glass bottle, and the submerged weight of the scaffold sample was recorded (n = 3 per group). The scaffold was then taken out, and the weight of the scaffold (containing cyclohexane in the void volume) was recorded. The porosity of the scaffold was calculated using the following equation:Porosity% = (M_w_ − M_d_)/(M_w_ − M_sub_) × 100(1)
where M_w_ is the cyclohexane-saturated scaffold, M_d_ is the dry mass of the scaffold, and M_sub_ is the submerged mass of the scaffold.

### 2.7. In Vitro Mineralization

The in vitro mineralization of the composite scaffolds in simulated body fluid (SBF, Gibco, Thermofisher, New York, NY, USA) was assayed. Scaffolds were immersed in SBF at 37 °C. The SBF was replaced every 3 days. At a pre-determined time, the samples were taken out of the SBF, gently rinsed with deionized water, and dried at 60 °C for 24 h. The surface morphology and composition of the scaffolds were characterized using FESEM and EDS. In addition, the concentrations of ions (Ca, P, Cu, and Si) after soaking in SBF were determined by inductively coupled plasma–atomic emission spectroscopy (ICP–AES, Agilent IC, Palo Alto, Santa Clara, CA, USA).

### 2.8. In Vitro Cytocompatibility

The scaffolds were autoclaved and placed in 24-well plates. Rat bone marrow stromal cells (rBMSCs) were obtained from the central laboratory at the Southern Medical University and cultured in Dulbecco’s Modified Eagle Medium (DMEM, Hyclone, Logan, UT, USA) supplemented with 10% fetal bovine serum (FBS, Hyclone, Logan, USA), and 1% penicillin/streptomycin (Pen/Strep, Gibco, Thermofisher, New York, NY, USA) in a humidified CO_2_ (5%) incubator at 37 °C. The medium was changed every two days during cell culture. After 1, 4, and 7 days of incubation, the proliferation of BMSCs on different scaffolds was performed using cell counting Kit-8 (CCK-8, Beyotime, Shanghai, China) according to the manufacturer’s instructions. The CCK-8 suspension cells were incubated for 2 h in 5% CO_2_ at 37 °C. Then, the absorbance of the solution at 450 nm was measured on a Synergy4 microplate reader (BioTek, Winooski, VT, USA).

The morphology of the cells on the scaffolds was observed by FESEM. Briefly, after culturing for 24 h, each sample was collected and fixed in 4% paraformaldehyde for 24 h. Then, the samples were dehydrated by gradient ethanol solution (10, 30, 50, 70, 90 and 100%) for 15 min, followed by air-drying. Finally, the scaffolds were sputter-coated with gold and observed under FESEM.

### 2.9. Animal Model

This study involved 24 5-month-old New Zealand white rabbits obtained from the Center of Experimental Animals, Southern Medical University (Guangzhou, China). All rabbits (male, body weight 2.0–3.0 kg) were obtained from the same animal holding facilities and were free from any eye disease. The rabbits were randomized into three groups, and each group comprised eight rabbits. The procedure was performed on one eye only. The rabbits were anesthetized with isoflurane inhalation, taking the side decubitus. When skin preparation and draping were completed, a wire eyelid speculum was applied. A 360° fornix-based conjunctival peritomy was performed at the limbus. Extraocular muscles were isolated and severed. The optic nerve was identified and then cut with enucleation scissors. The globe was completely removed. The anophthalmus model was built in all of the 24 rabbits, and then the orbital implant was carried out using sphere composite scaffolds (Φ = 12 mm). (The operation process is shown in Appendix A of the Appendix A) The rabbits were randomized into 3 groups: Group A (CP0), PEEK scaffolds; Group B (CP20), 20% CuBG/PEEK composite scaffolds; and Group C (CP40), 40% CuBG/PEEK composite scaffolds. After the scaffolds were implanted in the socket, the fascia and conjunctiva were sutured with 5-0 threads. Postoperative antibiotic ointment was used in the conjunctival sac for 5 days.

The three groups of rabbits were kept in different cages in the SPF laboratory. The presence of eye infection, implant extrusion or migration, ocular motility, and any evidence of wound breakdown were examined every week. In addition to two cases of material extrusion in Group A at 2 weeks, there was no material pull-out, migration, or incision infection during the period of feeding. (The postoperative situation is shown in Appendix A of the Appendix A.) The protocols of the animal test were approved by the Southern Medical University Experimental Animal Ethics Committee (NFYY-2019-73) and carried out in accordance with the institutional guidelines. All surgical procedures for evisceration and orbital implantation were conducted by a single surgeon and were required to follow standard ophthalmic surgical procedures.

### 2.10. Histochemical Staining

Animals were sacrificed by air embolization at the end of 4 weeks and 12 weeks. After that, enucleation with histopathological assessment was done to determine the presence of fibrovascular ingrowth and the rate of inflammatory reaction. The orbital implant of each group was fixed in 4% paraformaldehyde for at least 24 h before the gross sectioning was performed. The horizontal section was performed using a sharp surgical blade in a sawing motion from back to front. The interior of the implant was examined. After that, the horizontal section of the implant was placed in 4% paraformaldehyde, decalcified with 10% EDTA for 3 weeks, embedded with paraffin, and sliced into 5-μm-thick transverse sections following the standard method. Hematoxylin and eosin (H&E) staining and Masson staining were performed at room temperature. The slices were examined on a Leica DM5000 B (Leica, Wetzlar, Germany) microscope. The sections were examined under the microscope and were evaluated for the rate of inflammation and presence of fibrovascular ingrowths within the orbital implant. Semi-quantitative expression experiments of collagen fiber were performed with an inverted microscope. Each tissue slice was randomly counted by 15 high-power fields (×100), and images were acquired. They were measured using an Image-Pro Plus 6.0 color image analysis system. The integrated absorbance value and image area (S) of the blue regions were measured, the ratio of the absorbance of each field of view to the image area was obtained, and the average value was taken.

### 2.11. Statistical Analysis

Normality and homoscedasticity tests were carried out before applying ANOVA tests. The Kolmogorov–Smirnov normality test was used to test for normality. The homoscedasticity of the variables was tested by Levene’s test. One-way ANOVA tests were used to detect differences between groups. A *p*-value of less than 0.05 (*p* < 0.05) was considered statistically significant. Data were analyzed using SPSS 22.0 statistical software (IBM, Armonk, New York, NY, USA) and presented as mean ± SD.

## 3. Results

### 3.1. Characterization of CuBG/PEEK

The composite CuBG/PEEK material samples are shown in Figure 1. The surface microstructure of the composite scaffolds was observed and analyzed by FESEM. The microscopic morphology of each sample is shown in Figure 2A–C. It can be seen that all 3 groups of samples have a distinct pore structure (pore size > 400 μm) and high porosity (>70%, seen in Table 1), which is consistent with the particle size of the porogen (NaCl) used in the preparation of the scaffolds. As seen in Figure 2D, besides the C and O peaks belonging to the PEEK, Ca, Si, and Cu peaks were observed in the composite scaffolds by SEM–EDS, confirming the incorporation of CuBG into PEEK.

Figure 3A shows the XRD of the 3 groups of scaffolds. The 3 diffraction peaks at 18°, 22°, and 28° were characteristic peaks of PEEK. After CuBG was added, the positions of the diffraction peaks of the composite scaffolds (CP20 and CP40) did not shift significantly, but the intensity gradually decreased due to the slight decrease in the intensity of PEEK caused by the incorporation of CuBG. The FTIR spectrum shows the vibration peaks of PEEK in all 3 groups of scaffolds in Figure 3B. In the composite scaffolds, the peak at 1093 cm^−1^ was attributed to the Si-O-Si of CuBG, indicating that the composite scaffolds contained both PEEK and CuBG.

The three groups of scaffolds were immersed in SBF solution to observe the surface morphology changes to characterize the in vitro biological activity of the materials. As shown in Figure 4A–C, the surface of the PEEK scaffolds was still smooth. However, worm-like substances were found on the surface of the composite scaffolds, and the amount of them increased with the increase in CuBG content. It can be seen in Figure 4D that the peaks of Ca and P elements could be detected from the EDS spectrum of the composite scaffolds. The results showed that the introduction of CuBG significantly promoted the apatite-formation ability of the scaffolds, showing excellent bioactivity.

Figure 5 shows the ion concentration (Ca, P, Si, Cu) of the solution of the CP40 material in the SBF solution up to 14 days. It can be seen that the concentration of Ca and P elements continued to decrease during the whole soaking period, while the Si and Cu elements showed an upward trend, which might be caused by the degradation of CuBG.

As shown in Figure 6A–C, the rBMSCs showed better cell expansion and pseudopod growth on the surface of the composite scaffolds than did the PEEK scaffolds. Figure 6D shows the results of CCK-8 analysis of rBMSCs cultured on 3 groups of scaffolds for 1, 4, and 7 days. The cell viability of each group of scaffolds gradually increased with the culture time, while at the same culture time point, cell viability increased with the increase in CuBG content in the scaffolds. The results demonstrated that the addition of CuBG could improve cell adhesion and proliferation as compared with PEEK scaffolds.

### 3.2. General Observation Property

The densities of the fibrous vascular growth on the surfaces of the 3 groups of implants at 4 weeks post-evisceration are shown in Figure 7. It can be seen that the surface of the scaffolds of Group A is relatively smooth, but different degrees of new granulation tissue can be seen in Group B and Group C. The granulation tissue was bright red, granular, soft, and moist to the naked eye. This indicates that the formation of granulation tissue is facilitated by the addition of CuBG. However, a higher percentage of CuBG causes a slight decrease in the intensity of the composite PEEK. The surface of the composite PEEK may fragment over time, so it can be seen from the figure that the volume of composite PEEK in the orbit may be reduced slightly. After incising the scaffolds, it was observed that the three groups of scaffolds had fibrous vascular ingrowth. Compared to Group A, more fibrous tissue was seen in Group B and Group C.

### 3.3. H&E Staining

In order to observe the interaction between the scaffolds and the host tissue, H&E staining was conducted with an inverted microscope. The results of the H&E staining showed that only a small number of cells and collagen fibers were distributed in Group A, and the components were mainly inflammatory cells and foreign-body giant cells, which were consistent with obvious inflammatory reactions and foreign-body reactions. The degree of the growth of the new tissue in Group B and Group C was significantly increased. The main component was the new granulation tissue, which consisted of new, thin-walled capillaries and proliferating fibroblasts with a small number of inflammatory cells. The degree of growth, the number of new endothelial cells, and the number of functionally active fibroblasts also increased in Groups B and C, which indicated a time- and concentration-dependent manner (Figure 8).

### 3.4. Masson Staining

Masson staining is used to dye fiber in tissues. The collagen fiber shows as blue, the muscle fiber as red, and the nucleus as blue-violet [25]. The results suggest that collagen fibers in Group A were only present at the edge of the scaffolds. The distribution of collagen fibers in Groups B and C was more extensive with increasing content and time. Compared with Group A, the number and activity of fibroblasts in Groups B and C increased significantly and were positively correlated with content and time (Figure 9).

At 4 weeks, the results showed that the total amount of collagen fiber in Groups B and C increased significantly and did so with the increase in CuBG content. At 12 weeks, the results showed that the total content of collagen fiber in Group A was lower, in line with the fact that collagen fiber only existed on the edge of the scaffolds. The collagen fiber content of Group B and Group C increased with time (Figure 10).

## 4. Discussion

Porous implants have been widely adopted by surgeons performing enucleation and evisceration since the late 1980s, using materials such as hydroxyapatite, high-density polyethylene, aluminum oxide, etc. [5,6] However, those materials have usually been accompanied by the risks of bacterial penetration and implant exposure [12]. Thus, the search for the ideal orbital implant for the anophthalmic socket continues to evolve. The ideal orbital implant must have good histocompatibility and should have minimal rates of migration, extrusion, exposure, and infection [13].

PEEK is considered an advanced biomaterial used in medical implants, but it is a biologically inert material, which has limited its extensive biomedical application. Therefore, improving the bioactivity of PEEK is a crucial challenge that must be solved to fully realize its potential benefits [13]. At present, surface modification or compositing with bioactive ingredients has been widely harnessed to improve the bioactivity of PEEK. BG has good bioactivity and biocompatibility; in addition, copper ions are involved in the angiogenesis process. The introduction of bioactive materials in PEEK can significantly improve the bioactivity and biocompatibility of materials, making them more suitable for orbital implants in the present work.

In this study, we prepared a series of CuGB/PEEK composites with different levels of CuGB content. Compounding CuBG with PEEK is a physical process that would not change any chemical structure of PEEK. In consequence, the proper biocompatibility of CuGB/PEEK composites is foreseeable. CuBG particles were well-dispersed in the PEEK matrix, which was confirmed by the XRD and FTIR results. FESEM and EDS were used to analyze the topography and elemental distribution features of the sample surface. It is obvious that the pure PEEK scaffolds display the smoothest morphology, and the CuGB/PEEK scaffolds possess a rougher surface with micron-sized features, which may be the CuBG particles or their aggregates. The results showed that impregnating CuGB into the PEEK matrix significantly altered the surface morphology of the scaffolds, and the possible presence of CuBG particles could consequently improve the bioactivity of PEEK.

Besides in vitro evaluation, in vivo tissue response to the CuBG/PEEK scaffolds is crucial to the success of implantation. This is an experimental and observational study on the composite scaffolds and their consequences as orbital implants and the histopathological changes that occur in experimental, eviscerated rabbits. The PEEK composite with different concentrations of CuBG was used as the experimental group, with a pure PEEK counterpart serving as the control group. There were interconnecting pores of about 500 μm in diameter in the CuGB/PEEK orbital implant. The pores allowed for vascular tissue ingrowth and anchoring to the ocular socket. The introduction of bioactive materials can significantly improve the biocompatibility of materials. In addition, copper can also promote angiogenesis. This histopathological study evaluated the scaffolds, which were removed at 4 weeks and 12 weeks after implantation in the enucleated sockets of rabbits. Except for the pure PEEK group, some of the CuBG/PEEK implants had achieved complete vascularization at 4 weeks after implantation, and by 12 weeks, all of the 40% CuBG/PEEK implants were completely vascularized. Histological evidence shows that CuGB/PEEK scaffolds have good biocompatibility in rabbit eyes. There was good fibrovascular ingrowth and minimal to moderate inflammatory reaction observed. In this study, we found that CuGB/PEEK scaffolds successfully show fibrovascular ingrowth between and within the micropores of CuGB/PEEK architecture. Observations did not show any sign of rejection throughout this study. This phenomenon proves that there is a biocompatible environment at the host. The surface roughness of the scaffolds would allow a more stable fixation when fibrovascular ingrowth has occurred.

## 5. Conclusions

This study was carried out to assess the biocompatibility of CuGB/PEEK as an orbital implant in rabbits. Besides that, the histopathological reactions towards CuGB/PEEK orbital implants were also determined. There was good fibrovascular ingrowth, and a minimal inflammatory reaction was observed, as well as histological evidence of fibrovascularization within the implants as early as 4 weeks. This indicates a low risk of rejection and extrusion. With the increase in CuBG content, biocompatibility was enhanced. Taken together, in vitro and in vivo experiments here have showcased that 40% CuBG/PEEK composite scaffolds had the strongest ability in angiogenesis and tissue repair, and it was the most suitable and effective candidate as the orbital implants. Therefore, this study demonstrates the feasibility and possibility of using CuGB/PEEK for orbital implants.

## Figures and Tables

**Figure 1 materials-15-04410-f001:**
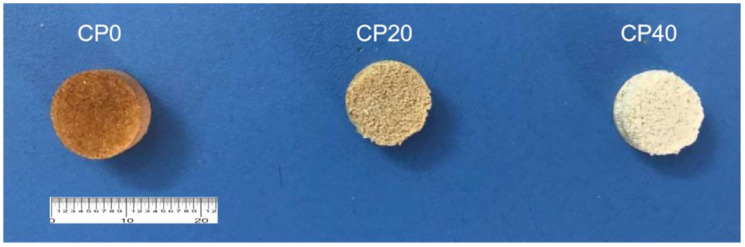
Digital photos of composite PEEK material samples (CP0, CP20, and CP40).

**Figure 2 materials-15-04410-f002:**
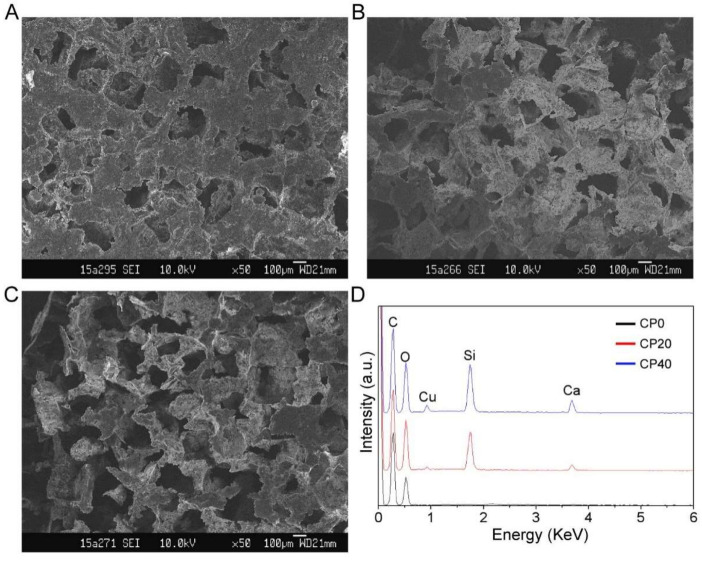
FESEM images of CP0 (**A**), CP20 (**B**), and CP40 (**C**), and EDS analysis (**D**) of the elemental composition of the scaffolds.

**Figure 3 materials-15-04410-f003:**
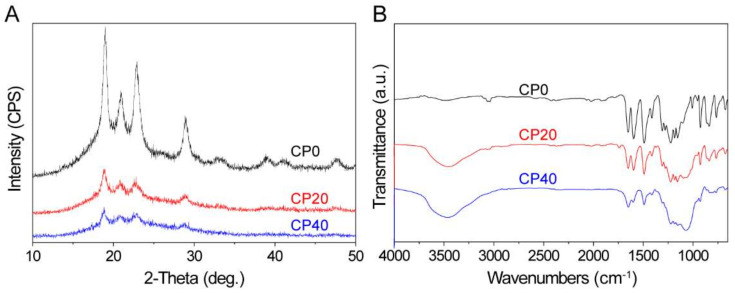
XRD (**A**) and FTIR (**B**) of CP0, CP20, and CP40.

**Figure 4 materials-15-04410-f004:**
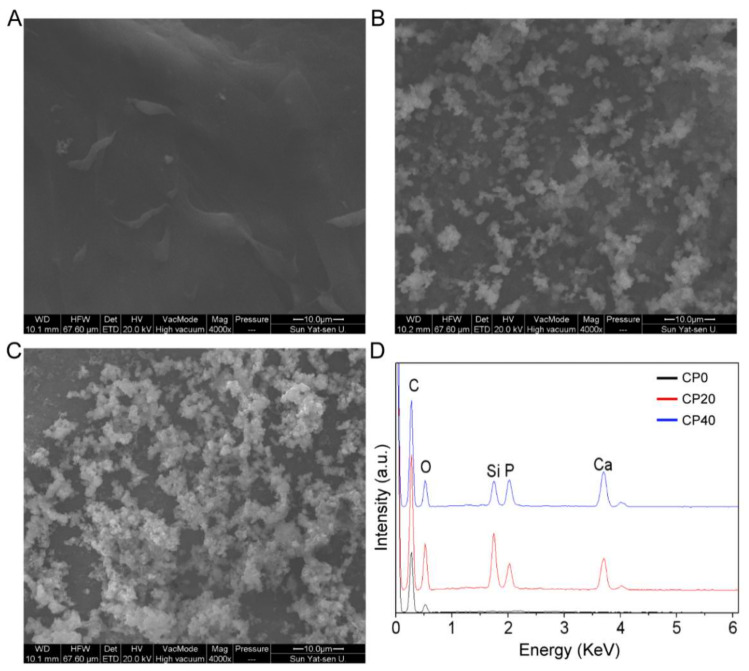
FESEM images of CP0 (**A**), CP20 (**B**), and CP40 (**C**) after immersion in SBF for 7 days, and EDS of scaffolds (**D**) after immersion.

**Figure 5 materials-15-04410-f005:**
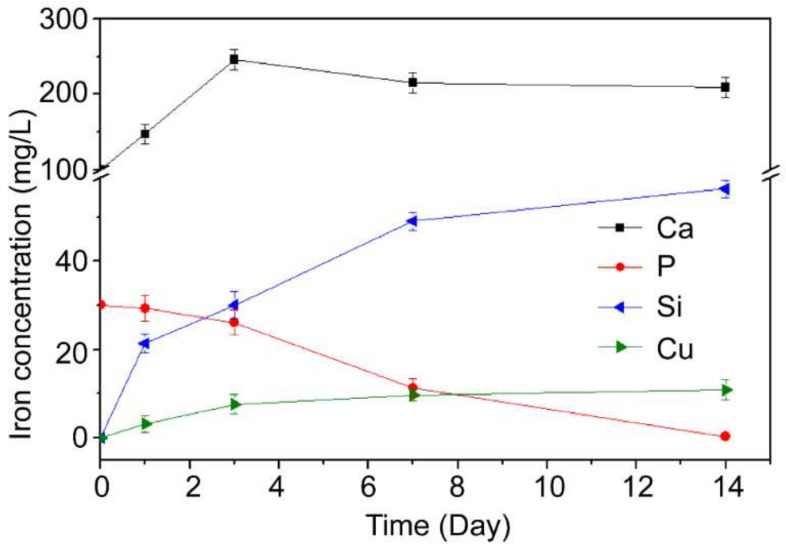
The changes of ion concentrations after 40% CuBG/PEEK (CP40) soaking in SBF solution for different amounts of time (means ± SD, *n* = 5).

**Figure 6 materials-15-04410-f006:**
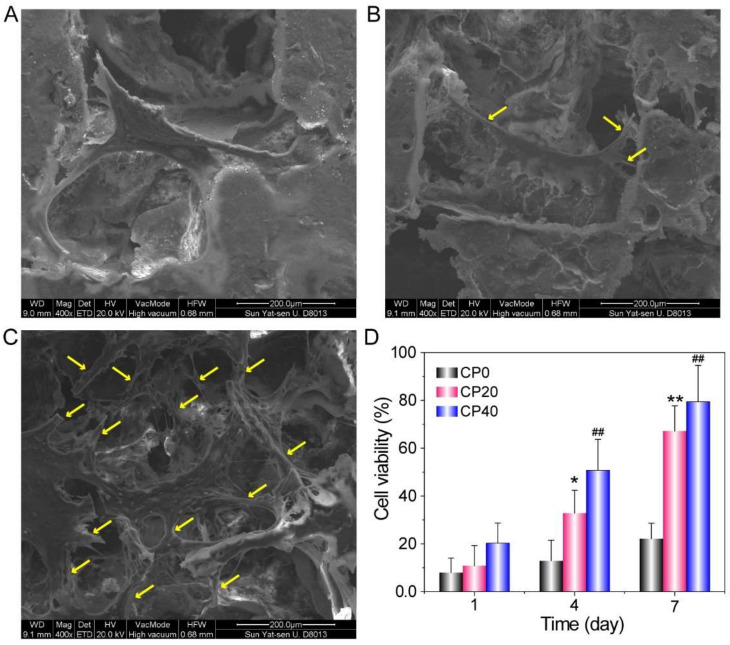
FESEM images of rBMSC cultured on CP0 (**A**), CP20 (**B**), and CP40 (**C**) at 24 h, and cell proliferation on the scaffold surfaces by the CCK-8 assay (**D**) (means ± SD, n = 5). The * *p* < 0.05 and ** *p* < 0.01 are the CP20 group vs. the CP0 group; ^##^
*p* < 0.01 are the CP40 group vs. the CP0 group at the same time.

**Figure 7 materials-15-04410-f007:**
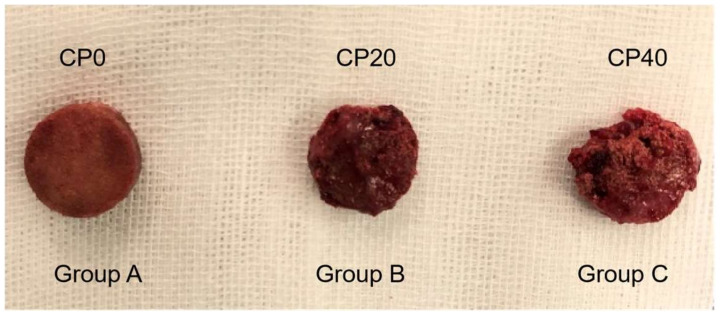
The density of fibrous vascular growth on the surface of 3 groups of implants at 4 weeks with CP0 (Group A), CP20 (Group B), and CP40 (Group C).

**Figure 8 materials-15-04410-f008:**
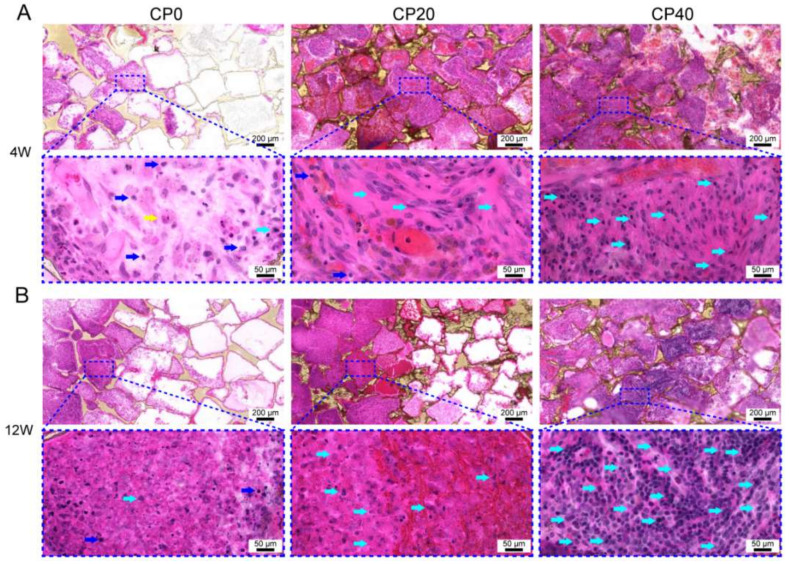
H&E staining of implant scaffolds at 4 weeks (**A**) and 12 weeks (**B**). Inflammatory cells (dark-blue arrows), giant cells (yellow arrow), and bone marrow-derived mesenchymal cells (light-blue arrows).

**Figure 9 materials-15-04410-f009:**
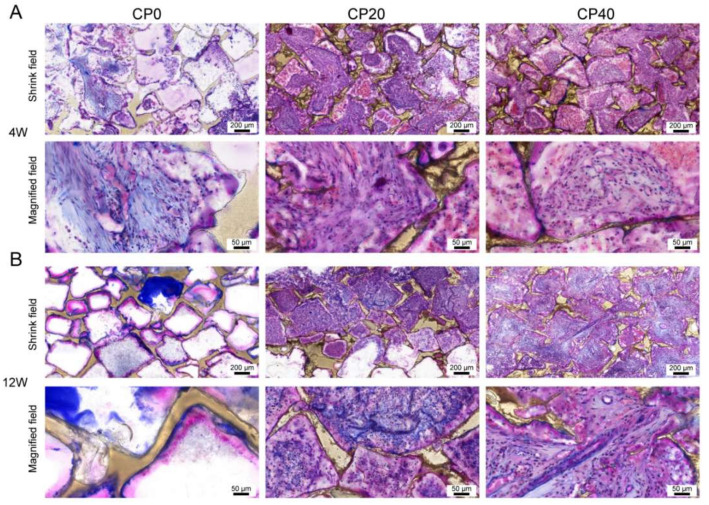
Masson staining of implant scaffolds at 4 weeks (**A**) and 12 weeks (**B**). The blue area represents collagen fiber, and the distribution of collagen fiber in Groups B and C was more extensive.

**Figure 10 materials-15-04410-f010:**
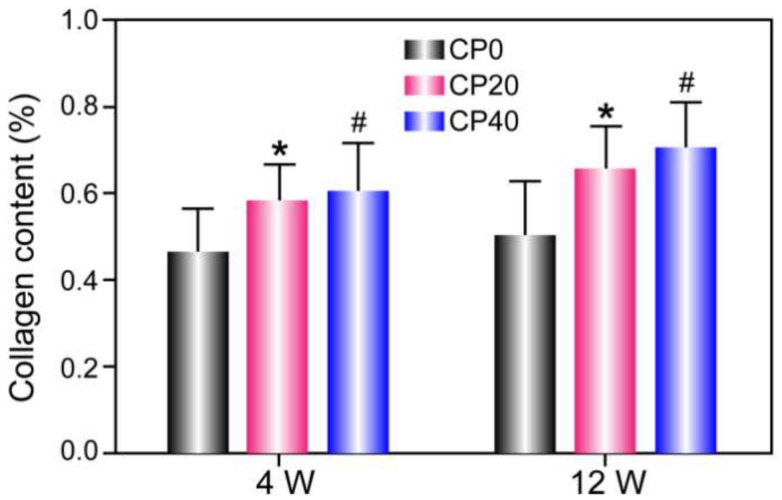
Semi-quantitative expression experiments of collagen fibers. The * *p* < 0.05 CP20 group vs. CP0 group and ^#^
*p* < 0.05 CP20 group vs. CP40 group. The values are represented as mean ± SD (n = 8).

**Table 1 materials-15-04410-t001:** The porosity of scaffold materials with different percentages of CuBG.

Samples	Porosity (%)	Pore size (μm)
PEEK (CP0)	72.1 ± 2.7	413.2 ± 7.6
20% CuBG/PEEK (CP20)	74.7 ± 3.1	435.9 ± 8.7
40% CuBG/PEEK (CP40)	75.3 ± 3.3	478.1 ± 6.1

## Data Availability

The data presented in the study are available on request from the corresponding author.

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
