# Peer review of "Copper-Doped Bioactive Glass/Poly (Ether-Ether-Ketone) Composite as an Orbital Enucleation Implant in a Rabbit Model: An In Vivo Study"

_materials, 2022, doi:10.3390/ma15134410_

Round 1

Reviewer 1 Report

Review of Copper-doped bioactive glasspoly(ether-ether-ketone) composite as orbital enucleation implant in rabbit model An in vivo study

REVIEWER’s OPINION:

ACCEPTED WITH MAYOR REVISION

Authors attempt characterize ex vivo the physical and biocompatibility properties (contact with BMSC) of copper-doped bioactive glass in poly(ether-ether-ketone) (CuBG/PEEK) composite scaffolds as orbital enucleation implant fabricated by cool-pressed sintering and particle leaching techniques in the biocompatibility and then performed an histopathology analysis in vivo.

 This study can elucidate the behavior of different orbital substitutes used in the ophthalmological reparation in a comparative manner. This could be a quantum leap in current clinical therapeutics for the treatment for reparative ophthalmologic treatments. However, there are many design and experimental development suggestion that could be applied to improve the study and reach the standards of quality subject to this type of article.

OVERALL COMMENTS:

Authors must be revised, is there one author left?

 Ke Xiong1, 3, Mengen Zhao1, Zhaoying Wu1, Wei Zhang2, *, Chao Zhang1 and XXX

Material and methods should clarify how many samples were used in each analysis. And perform the statistical comparisons between all the groups.

The election of the statistical analysis used should be explained. See details in material and methods section.

Statistical comparisons in figures should be clarified as figure 6.  

Overall comparative data presented by the authors could be used to reach an enhanced conclusion that provides to the reader clearer information about the comparative study performed (which is the best treatment regarding the results obtained?) that can be included in the abstract.  

Some figures must be modified. See details in results section.

INTRODUCTION:

  • It is the reviewer opinion that the part of the introduction where it is mentioned “synthesized by Friedel-Crafts polycondensation“ the font size must be revised.
    • References should be used after the sentences “As a high-performance polymer, PEEK is known for its excellent mechanical properties as well as extraordinary thermal stability and chemical resistance against oils, acids, and biological fluids” and “but also induce the proliferation of endothelial cells, and blood vessel formation mainly depends on the activity of endothelial cells”.  

MATERIALS AND METHODS:

  • Histology is considered one of the most useful and accurate quality controls in tissue engineering, including the assessment of differentiation.
  • It is the reviewer opinion that the morphology of the cells on the scaffolds should have been observed by FESEM in the middle or last timepoint. Why did you choose the first timepoint to determine the cytocompatibility test?.
  • Statistical analysis should be explained. To apply ANOVA (or other parametric test) is highly recommended that authors should perform first normality test (for example Shapiro-Wilk or Kolmogorov-Smirnov) and homogeneity of variance or homoscedasticity test (such as Levene test). In case of these necessary criteria are not met, the statistical tests to be applied should be non-parametric. The authors should explain why they elected this method to compare the different groups (n=?? each) and add references about if this method is used correctly with the number of samples used.

RESULTS:

  • It is the reviewer opinion that the acronyms as CT have to be stablished from the very outset.
  • Authors should clarify and explain the different significant comparisons performed in the figures. The modification of the legend of figure 10 “The *p<05 and #p<0.05 vs CP0 group” clarifying which letter corresponds to which comparison is highly recommended.
  • In the section “Characterization of CuBG/PEEK” authors affirm that “The cell viability of each group scaffolds gradually increased with the culture time. While at the same culture time point, the cell viability increased with the increasing of CuBG content in scaffolds.” It is significant the difference between the different timepoints?
  • In the section “Histological examination” authors affirm that “were mainly inflammatory cells and foreign body giant cells”. It is the reviewer opinion that is not possible to identify the inflammatory cells and giant cells at these magnifications, so images should be enhanced and mark with arrows or asterisks the different structures that the authors referred to.
  • It is the reviewer opinion that “Semi-quantitative expression experiments of collagen fiber were performed with an inverted microscope. Each tissue slice was randomly counted for 15 high power fields (×100), and images were acquired. It was measured using Image-Pro Plus 6.0 colour image analysis system. The integrated absorbance value and image area (S) of the blue region were measured, and the ratio of the absorbance of each field of view to the image area was obtained, and the average value was taken” should be located in material and methods section.

DISCUSSION:

  • Discussion must be rewritten, enhanced and extended. In this case, authors only commented their own results without discussing, hypothesizing or comparing to the literature. All presented results should be discussed and compared with bibliographic references (of other materials or solution used) whenever possible or hypothesize the reason to explain the results.
  • It is the reviewer opinion that the main point of this study is to highlight the differences between the different strategies used. Due to this fact, authors should elaborate a conclusion that summarizes the different effects observed in the groups and observing its characteristics explaining which substitute is recommended to be used in the citated pathology.

Author Response

Point 1:  Authors must be revised, is there one author left?

Response 1: Thanks for pointing out this mistake, we did miss one co-author when uploading the manuscript. Actually, there are five co-authors, including Ke Xiong, Mengen Zhao, Zhaoying Wu, Wei Zhang and Chao Zhang. We have now updated the information of the co-authors in the submission system.

Point 2:  Material and methods should clarify how many samples were used in each analysis. And perform the statistical comparisons between all the groups.

Response 2: Thanks for the valuable adviced. We have accordingly clarified the sample numbers and conducted statistical analysis on corresponding data.

Point 3:  The election of the statistical analysis used should be explained. See details in material and methods section.

Response 3: Thanks for the suggestions, please see our explanation in Response 11.

Point 4:  Statistical comparisons in figures should be clarified as Figure 6.  

Response 4: Thanks for the suggestion. Corresponding corrections have been made in Figure 6. The changes are as follows: The *p<0.05 and **p<0.01 is CP20 vs CP0 group; the #p<0.05 and ##p<0.01 is CP40 vs CP0 group at the same time.

Point 5:  Overall comparative data presented by the authors could be used to reach an enhanced conclusion that provides to the reader clearer information about the comparative study performed (which is the best treatment regarding the results obtained?) that can be included in the abstract.  

Response 5: Thanks for the helpful comment, we have modified our stastical comparisons (the detailed correction are contained in figure legends). From all of the comparasion data in the manuscript, we draw the conclusion that the CP40 group had the strongest ability in bone formation and tissue repair.

Point 6:  Some figures must be modified. See details in results section.

Response 6: Thanks for the suggestion, figures in the manuscript have been checked thoroughly and modified/updated accordingly.

Point 7:  It is the reviewer opinion that the part of the introduction where it is mentioned “synthesized by Friedel-Crafts polycondensation“ the font size must be revised.

Response 7: Thanks for the suggestions, we have revised the font size to Italic.

Point 8:  References should be used after the sentences “As a high-performance polymer, PEEK is known for its excellent mechanical properties as well as extraordinary thermal stability and chemical resistance against oils, acids, and biological fluids” and “but also induce the proliferation of endothelial cells, and blood vessel formation mainly depends on the activity of endothelial cells”.

Response 8: Thanks for the suggestion, related references have been added at the end of the sentences.

Point 9:  Histology is considered one of the most useful and accurate quality controls in tissue engineering, including the assessment of differentiation.

Response 9: We totally agree with the reviewer. Since there have been several related reports, we are more interested in the detailed in vitro assessment of the composite scaffold as bone healing/regeneration material in this work; we hope our efforts may help researchers in this field gain enough insights on this topic.

Point 10:  It is the reviewer opinion that the morphology of the cells on the scaffolds should have been observed by FESEM in the middle or last time point. Why did you choose the first time point to determine the cytocompatibility test?

Response 10: Thanks for the suggestions, because the culture of cell is the most vulnerable in the first 24 hours, the compatibility and adhesion of materials to cells can be accurately reflected. The differences of each group can be seen from the FESEM, and 40% CuBG/PEEK group is the best.

Point 11:  Statistical analysis should be explained. To apply ANOVA (or other parametric test) is highly recommended that authors should perform first normality test (for example Shapiro-Wilk or Kolmogorov-Smirnov) and homogeneity of variance or homoscedasticity test (such as Levene test). In case of these necessary criteria are not met, the statistical tests to be applied should be non-parametric. The authors should explain why they elected this method to compare the different groups (n=?? each) and add references about if this method is used correctly with the number of samples used.

Response 11: Thanks for the valuable suggestions, we have added in statistical analysis section. Normality and homoscedasticity tests were carried out before applying ANOVA tests. The Kolmogorov–Smirnov normality test was used to test for normality. The homoscedasticity of the variables was tested by Levene’s test. Sample size was not predetermined but number of samples are consistent with previous publications.

Point 12:  It is the reviewer opinion that the acronyms as CT have to be stablished from the very outset.

Response 12: Thanks for the comment, CT is the abbreviation for computed tomography, we have accordingly added the full name in the Introduction section.

Point 13:  Authors should clarify and explain the different significant comparisons performed in the figures. The modification of the legend of figure 10 “The *p<05 and #p<0.05 vs CP0 group” clarifying which letter corresponds to which comparison is highly recommended.

Response 13: Thanks for the suggestion, we have modified the legend of figure 10, “The *p<0.05 CP20 group vs CP0 group and #p<0.05 CP20 group vs CP40 group.”

Point 14:  In the section “Characterization of CuBG/PEEK” authors affirm that “The cell viability of each group scaffolds gradually increased with the culture time. While at the same culture time point, the cell viability increased with the increasing of CuBG content in scaffolds.” It is significant the difference between the different timepoints?

Response 14: Thanks for the comment, we have re-analyzed the data and found that cell activity increased gradually with the increasing of CuBG content and the time. And the description was added in the Figure 6.

Point 15: In the section “Histological examination” authors affirm that “were mainly inflammatory cells and foreign body giant cells”. It is the reviewer opinion that is not possible to identify the inflammatory cells and giant cells at these magnifications, so images should be enhanced and mark with arrows or asterisks the different structures that the authors referred to.

Response 15: Thanks for the helpful advice, we have modified Figure 8 and marked inflammatory cells, giant cells, and bone marrow derived mesenchymal cells with different colored arrows. 

Point 16: It is the reviewer opinion that “Semi-quantitative expression experiments of collagen fiber were performed with an inverted microscope. Each tissue slice was randomly counted for 15 high power fields (×100), and images were acquired. It was measured using Image-Pro Plus 6.0 colour image analysis system. The integrated absorbance value and image area (S) of the blue region were measured, and the ratio of the absorbance of each field of view to the image area was obtained, and the average value was taken” should be located in material and methods section.

Response 16:  Thanks for your helpful advice, we have moved the related description to the Material and Methods section.

Point 17:  Discussion must be rewritten, enhanced and extended. In this case, authors only commented their own results without discussing, hypothesizing or comparing to the literature. All presented results should be discussed and compared with bibliographic references (of other materials or solution used) whenever possible or hypothesize the reason to explain the results.

Response 17: Thanks for the valuable advice. Discussion already has been re-written with enhanced and extended discussion.

Point 18:  It is the reviewer opinion that the main point of this study is to highlight the differences between the different strategies used. Due to this fact, authors should elaborate a conclusion that summarizes the different effects observed in the groups and observing its characteristics explaining which substitute is recommended to be used in the citated pathology.

Response 18: Thanks for the advice. The conclusion we can draw from the study is that 40%CuGB/PEEK was the most suitable and effective as the orbital implant. The study demonstrates the feasibility and possibility of using CuGB/PEEK for orbital implant. 

Reviewer 2 Report

In the present study “Copper-doped bioactive glass/poly(ether-ether-ketone) composite as orbital enucleation implant in rabbit model: An in vivo study”, we aim to design a PEEK based implant material with enhanced bioactivity and evaluate its applicability as orbital implant. To this end, copper-doped bioactive glass nanoparticles (CuBG) were prepared and incorporated into the PEEK matrix to fabricate the CuBG/PEEK composite scaffolds; the scaffolds were implanted into an experimental eviscerated rabbit to observe the consequences and histopathological changes after implantation.

The manuscript looks interesting, but some modifications must be made before publication. My comments could be found below:

  • Authors must correct some grammar mistakes: e.g. bioacticvity , nanoaprticles,
  • Authors mention that the PEEK powder and CuBG powder were mixed at a certain ratio (80:20, 60:40). A minimal reference or a discussion about the selection of these ratios must be made.
  • In section 2. Materials and Methods, must be inserted a subsection Characterization of CuBG/PEEK scaffolds. Also, the description of the characterization methods must include the detailed purpose of this analysis not just. In this section, I expect to find some details about porosity measurements.
  • Regarding the figure 2. FESEM images of CP0 (A), CP20 (B) and CP40 (C), and EDS of the scaffolds (D), please, mention the point where EDS was made on a SEM image.
  • Regarding the porosity, author could explain how was determined. A discussion about the thickness of the walls could be made in conjunction with the CuBG ratio because according to the SEM images looks to be a difference.
  • I think that a higher percent of CuBG induce a decrease of pore wall and the scaffolds will lose faster the interconnectivity of the pores. Maybe an FESEM analysis of the samples shown in figure 7 could confirm or not this supposition.
  • Authors mention that the incorporation of CuBG will decrease the crystallinity of PEEK. In this case, I think that the implant disintegration, fragmentation and partial extrusion will bring some irreversible side effects. Anyway, a minimal discussion about this fact must appear.
  • Authors perform SEM analysis on the scaffolds after immersion in SBF solution to observe the surface morphology changes. I would like to see a Sem image at 50x, in order to compare with the original scaffolds (shown in figure 3). For example, in figure 4a is shown just a detail, without any porosity aspect.
  • The description of the figure 8 and figure 9 must be reformulated (could use the description of figure 6).
  • According to the image shown in figure 7, a higher percent of CuBG could induce the modification of the volume in the orbit over time. Looking forward to finding some discussion about this.
  • I quite well-known the complexity of surgical procedure of implantation is limitative. A comparative discussion about this new proposed implant and other commercial implants must be made in term of surgical procedure.

Author Response

Dear Editors and Reviewers,

On behalf of my co-authors, we thank you very much for giving us an opportunity to revise our manuscript. We would like to express our gratitude for the valuable comments and suggestions from the editors and reviewers on our manuscript (Title: Copper-doped bioactive glass/poly(ether-ether-ketone) composite as orbital enucleation implant in a rabbit model: An in vivo study). We have taken all the comments into consideration and have made careful revisions to the manuscript accordingly and every revised position was marked using blue color in the revised manuscript. Below are our responses to the Reviewer's comments. The reviewers’ comments are shown in black font, and our answers are shown in red font.

Response to Reviewer 2 Comments

Point 1:  Authors must correct some grammar mistakes: e.g. bioacticvity , nanoaprticles,

Response 1: Thanks for the suggestion, the manuscript has been checked thoroughly and modified/updated accordingly.

Point 2:  Authors mention that the PEEK powder and CuBG powder were mixed at a certain ratio (80:20, 60:40). A minimal reference or a discussion about the selection of these ratios must be made. 

Response 2:  Thanks for the valuable suggestions, the mixing of PEEK powder and CuBG powder in a certain ratio is carried out with reference to the previous literature (see DOI: 10.1039/c7tb02344h).

Point 3:  In section 2. Materials and Methods, must be inserted a subsection Characterization of CuBG/PEEK scaffolds. Also, the description of the characterization methods must include the detailed purpose of this analysis not just. In this section, I expect to find some details about porosity measurements.

Response 3: Thanks for the advice. We have added Porosity measurement in the Materials and Methods section, “Insert content: Porosity measurement: the porosity (P) of theCuBG/PEEK composite scaffolds was determined according to Archimedes principle using a gravity bottle. Briefly, the dry mass of the scaffold was measured, followed by soaking the scaffold in cyclohexane in a gravity bottle and recording the submerged weight of the scaffold sample. The scaffold was then taken out and the weight of the scaffold (containing cyclohexane in the void volume) was recorded. The porosity of the scaffold was calculated following the equation below: Porosity% = (Mw-Md) / (Mw-Msub) ×100 where Mw is cyclohexane saturated scaffold. Md is dry mass of the scaffold and Msub is submerged mass of the scaffold.”, and have inserted subsections.

Point 4:  Regarding the figure 2. FESEM images of CP0 (A), CP20 (B) and CP40 (C), and EDS of the scaffolds (D), please, mention the point where EDS was made on a SEM image.

Response 4: Thanks for the comment. We have revised Figure 2 accordingly. (EDS analysis of the elemental composition of the scaffolds. Besides the C and O peaks belonging to PEEK, Ca, Si and Cu peaks were observed in the composite scaffolds)

Point 5:  Regarding the porosity, author could explain how was determined. A discussion about the thickness of the walls could be made in conjunction with the CuBG ratio because according to the SEM images looks to be a difference.

Response 5: Thanks for the helpful advice, we have added porosity measurements in the Materials and Methods section. From SEM images and actual measurement, there is little difference in porosity among the three groups. The main factor affecting the material is the bioactive materials rather than the porosity.

Point 6: I think that a higher percent of CuBG induce a decrease of pore wall and the scaffolds will lose faster the interconnectivity of the pores. Maybe an FESEM analysis of the samples shown in figure 7 could confirm or not this supposition.

Response 6: Thanks for the comment. All three groups of samples have a distinct pore structure and high porosity, and not much has changed between them. Figure 7 showed the density of fibrous vascular growth on the surface of three groups of implants, affected by the percent of CuBG.

Point 7:  Authors mention that the incorporation of CuBG will decrease the crystallinity of PEEK. In this case, I think that the implant disintegration, fragmentation and partial extrusion will bring some irreversible side effects. Anyway, a minimal discussion about this fact must appear.

Response 7: Thanks for the suggestion, the intensity of PEEK peak in the XRD spectrum decreases slightly with the incorporation of CuBG.  The surface of the materials may be disintegrated or fragmented.  This can also happen with commercial implants, such as hydroxyapatite implants. Due to the incorporation of active materials and copper ions, the composite scaffolds exhibited the strongest ability in angiogenesis and tissue repair.  The side effects may be negligible in this study as evidenced by the histochemical staining.  

Point 8:  Authors perform SEM analysis on the scaffolds after immersion in SBF solution to observe the surface morphology changes. I would like to see a SEM image at 50x, in order to compare with the original scaffolds (shown in figure 3). For example, in figure 4a is shown just a detail, without any porosity aspect.

Response 8: Thanks for the suggestion, from Fig 4, SEM analysis on the scaffolds after immersion in SBF solution to observe the surface morphology changes. The results showed that the introduction of CuBG significantly promoted the apatite-formation ability of scaffolds, showing excellent bioactivity.

Point 9:  The description of the figure 8 and figure 9 must be reformulated (could use the description of figure 6).

Response 9: Thanks for the suggestion. Corresponding corrections have been made in Figure 8 and Figure 9. ï¼ˆFigure 8 H&E staining of implant scaffolds in 4 weeks(A) and 12 weeks(B). Inflammatory cells (dark blue arrow), giant cells (yellow arrow ), and bone marrow-derived mesenchymal cells (light blue arrow). Figure 9 Masson staining of implant scaffolds in 4 weeks(A) and 12 weeks(B). The blue area represents collagen fiber, and the distribution of collagen fiber in group B and group C was more extensive.)

Point 10: According to the image shown in figure 7, a higher percent of CuBG could induce the modification of the volume in the orbit over time. Looking forward to finding some discussion about this.

Response 10:  Thanks for the suggestion, a higher percent of CuBG cause a slight decrease the intensity of the composite PEEK. The surface of the composite PEEK may fragmentate over time, so it can be seen from Figure 7 that the volume of composite PEEK in the orbit may be reduced slightly. We have accordingly added the discussion about this in the manuscript.

Point 11:  I quite well-known the complexity of surgical procedure of implantation is limitative. A comparative discussion about this new proposed implant and other commercial implants must be made in term of surgical procedure.

Response 11: Thanks for the suggestion. Many commercial implants, such as hydroxyapatite, high-density polyethylene, aluminum oxide, and bone cement, have been used in clinical practice. A number of problems such as the risk of infection, development of late exposures, and formation of pyogenic granuloma remain unsolved. This new proposed implant possesses a similar density/weight to the natural globe, proper porosity, appreciable histocompatibility, and achieves minimal rates of infection or exposure.

Round 2

Reviewer 1 Report

Dear authors, 

It is the reviewer opinion that after all the changes performed the manuscript has enhanced in a qualitative way but there are some critical points that were not covered:

  1. Point 7: Font size were not changed
  2. Point 10: The reviewer does not agree with this response, 24 h are not enough to determine the biocompatibility of one biomaterial that it is supposed to be permanent in a future patient. Also, authors should be consistent with the time points used (as figure 4, 7 days).
  3. Point 11: Authors did not provide essential information about number of samples used in each experiment. The author´s response was "Sample size was not predetermined but number of samples are consistent with previous publications" This fact could compromise the accuracy and reproducibility of this study. 
  4. Point 15: It is the reviewer opinion that the conclusions obtained after the histological assesment are wrong. Firstly, gigant cells marked have not the typical size and morphology of this kind of cells (compare the size of this marked cells with the others). Secondly, in this figure there are no differences between the inflamatory cells of  CP20 at 4 weeks and CP40 at 12 weeks. Thirdly, in order to identify the different type of cells, specific markers should have been used for each kind of cells (especially BMSC).

In this context, it is the reviewer opinion that the article does not meet the enough quality standards for the publication in this magazine. The reviewer´s opinion is REJECT AND RESUBMIT.  

Author Response

Dear Editors and Reviewers,

On behalf of my co-authors, we thank you again for your kindness and valuable comments and suggestions on our manuscript (Title: Copper-doped bioactive glass/poly(ether-ether-ketone) composite as orbital enucleation implant in a rabbit model: An in vivo study). We have taken all the comments into consideration and have made careful revisions to the manuscript accordingly and every revised position was marked using blue color in the revised manuscript. Please kindly check attached Responses to the Reviewer's Comments for evaluation.

Response to Reviewer 1 Comments

Point 7: Font size were not changed.

Response: Thanks for pointing out this mistake, we have reviewed the manuscript and modified the font size.  

Point 10: The reviewer does not agree with this response, 24 h are not enough to determine the biocompatibility of one biomaterial that it is supposed to be permanent in a future patient. Also, authors should be consistent with the time points used (as figure 4, 7 days).

Response: Thanks for the valuable suggestion, we totally agree with the reviewer on the cyto-/bio-compatibility of long-term implants. Actually, the SEM images (Figure 6A-C) of cells on the scaffolds after 24 hour of seeding would only be useful to evaluate the cell attachment and adhesion. To accurately demonstrate the cytocompatibility of scaffolds towards rBMSCs, a 7-day assay was performed (Figure 6D) with the assistance of CCK-8 kit. The MTT result demonstrated that the CP20 and CP40 groups showed significant proliferation compared with control group at 4 and 7 days.

Point 11: Authors did not provide essential information about number of samples used in each experiment. The author´s response was "Sample size was not predetermined but number of samples are consistent with previous publications" This fact could compromise the accuracy and reproducibility of this study. 

Response: Thanks for the valuable suggestion. We have checked the manuscript and added the number of samples in each experiment. The sample size estimation requires mean, standard deviation, type 1 error, and statistical efficiency. This study was a randomized controlled study to evaluate the composite material as orbital enucleation implant, and the research object was the rabbit. The sample sizes were obtained based on pre-experiments and parameters in the published literature (DOI:10.4103/0976-500X.119726). If the sample size is too small then adding more animals will increase the chance of getting more significant result, but if it is too large then adding more animals will not increase the chance of getting significant results. Though, this method is based on ANOVA, it is applicable to all animal experiments. The sample size should be considered as an adequate.

Point 15: It is the reviewer opinion that the conclusions obtained after the histological assesment are wrong. Firstly, gigant cells marked have not the typical size and morphology of this kind of cells (compare the size of this marked cells with the others). Secondly, in this figure there are no differences between the inflamatory cells of CP20 at 4 weeks and CP40 at 12 weeks. Thirdly, in order to identify the different type of cells, specific markers should have been used for each kind of cells (especially BMSC).

Response: Thanks for the valuable suggestion. Figure 8 have been checked thoroughly and modified/updated under the guidance of a pathologist. In fact, there are no statistical difference between the inflammatory cells of CP20 at 4 weeks and CP40 at 12 weeks. In this study, we used hard tissue sections that could not be labeled using immunofluorescence or immunohistochemical techniques. We worked with pathologists to distinguish cells by cell size and morphology.

Reviewer 2 Report

The manuscript could be published. 

Author Response

Thank you for your valuable advices, which are very helpful to improve the quality of our manuscript.

This manuscript is a resubmission of an earlier submission. The following is a list of the peer review reports and author responses from that submission.